# A Systematic Review and Meta-Analysis of MicroRNA as Predictive Biomarkers of Acute Kidney Injury

**DOI:** 10.3390/biomedicines12081695

**Published:** 2024-07-30

**Authors:** Naomi Brown, Marius Roman, Douglas Miller, Gavin Murphy, Marcin J. Woźniak

**Affiliations:** Department of Cardiovascular Sciences and NIHR Cardiovascular Biomedical Research Unit, Glenfield Hospital, University of Leicester, Leicester LE3 9QP, UK; nab40@leicester.ac.uk (N.B.); marius.roman@leicester.ac.uk (M.R.); douglas.miller@nhs.net (D.M.); gjm19@leicester.ac.uk (G.M.)

**Keywords:** microRNAs, acute kidney injury, biomarkers, surgical procedure, sepsis

## Abstract

Acute kidney injury (AKI) affects 10–15% of hospitalised patients and arises after severe infections, major surgeries, or exposure to nephrotoxic drugs. AKI diagnosis based on creatinine level changes lacks specificity and may be delayed. MicroRNAs are short non-coding RNA secreted by all cells. This review of studies measuring miRNAs in AKI aimed to verify miRNAs as diagnostic markers. The study included data from patients diagnosed with AKI due to sepsis, ischaemia, nephrotoxins, radiocontrast, shock, trauma, and cardiopulmonary bypass. Out of 71 studies, the majority focused on AKI in sepsis patients, followed by cardiac surgery patients, ICU patients, and individuals receiving nephrotoxic agents or experiencing ischaemia. Studies that used untargeted assays found 856 differentially regulated miRNAs, although none of these were confirmed by more than one study. Moreover, 68 studies measured miRNAs by qRT-PCR, and 2 studies reported downregulation of miR-495-3p and miR-370-3p in AKI patients with sepsis after the AKI diagnosis. In three studies, upregulation of miR-21 was reported at the time of the AKI diagnosis with a significant pooled effect of 0.56. MiR-21 was also measured 19–24 h after cardiac surgery in three studies. However, the pooled effect was not significant. Despite the considerable research into miRNA in AKI, there is a knowledge gap in their applicability as diagnostic markers of AKI in humans.

## 1. Introduction

Acute kidney injury (AKI) is a significant public health issue that occurs in up to 50% of critically ill patients [1]. In the United Kingdom, it is estimated that AKI affects approximately 20% of hospital admissions, translating to around 100,000 deaths annually due to the condition or its complications [2]. In Europe, AKI affects around 13% of hospitalised patients, with over 50% incidence in intensive care units (ICUs) [3]. In the United States, AKI in hospital inpatients ranges from 12% (injuries and poisoning) to 20% (cardiac surgery patients) and increases mortality rates 6.5-fold [4].

Patients who develop AKI have an increased risk of mortality and are predisposed to adverse events following recovery. AKI is characterised by a rapid loss of kidney function and can arise from various aetiologies, including sepsis, nephrotoxicity, or major surgery [5]. AKI following cardiac surgery occurs in up to 30% of patients, and it is associated with increased postoperative complications and mortality rates [6,7]. In patients with sepsis, AKI occurs in 43% of cases, exacerbating the severity of the condition and complicating treatment strategies [8]. Exposure to nephrotoxic agents, such as certain antibiotics and chemotherapeutic drugs, is another significant cause of AKI, affecting up to 20% of patients receiving these treatments [9,10,11,12].

The economic impact of AKI is substantial, with increased healthcare costs attributed to prolonged hospital stays, the necessity for renal replacement therapy, and the management of associated complications [13]. For instance, in the UK, the annual cost of AKI is estimated to be between GBP 434 million and GBP 620 million [14]. Moreover, AKI is a risk factor for the development of chronic kidney disease (CKD), which further complicates outcomes and increases the likelihood of patients requiring long-term dialysis or kidney transplantation [15]. The progression from AKI to CKD is observed in 30% of AKI survivors, highlighting the long-term impact of this acute condition on kidney health [16].

Early detection and progress staging of AKI are imperative for effective clinical intervention. However, current diagnostic markers, such as serum creatinine and urine output, lack sensitivity and specificity, resulting in inaccurate and delayed diagnoses of AKI [17]. Serum creatinine, the most commonly used marker, does not rise until substantial kidney damage has occurred, often lagging behind actual injury by 24 to 48 h [18]. Similarly, urine output measurements can be influenced by numerous factors unrelated to kidney function, further complicating timely diagnosis [19]. This diagnostic challenge underscores the urgent need for more reliable biomarkers that can detect AKI at an earlier stage, allowing for prompt and targeted therapeutic interventions.

In recent years, intensive research has focused on developing new and improved AKI biomarkers. Specifically, the urinary markers TIMP2 and IGFBP7 have shown efficacy in predicting the risk of AKI in cardiac surgery patients [20], whilst the renal tubular stress marker Dickkopf has been identified as a novel target for early diagnosis [21]. These biomarkers hold promise for enhancing early detection and predicting outcomes in AKI patients. Nevertheless, challenges remain in translating novel biomarkers into clinical practice, primarily due to complexities in interpretation, specificity issues, and the need for comprehensive evaluation [22].

Micro-RNAs (miRNAs) are a class of non-coding RNA oligonucleotides that play vital roles in posttranslational gene expression and intercellular signalling. miRNAs regulate gene expression by targeting specific regions in the 3′-untranslated regions (3′-UTRs) of messenger RNA (mRNA), which can lead to either the inhibition of mRNA translation or its degradation [23]. In humans, more than 2500 mature miRNAs have been discovered, each implicated in a diverse array of biological processes crucial for normal cellular function and development [24]. Several of these miRNAs have been found to play roles in the pathogenesis of AKI [25,26]. Indeed, the use of miRNAs as therapeutic agents has been thoroughly evaluated in pre-clinical models of AKI [27]. However, further evaluation is necessary to validate their specificity, sensitivity, and reproducibility before miRNAs can be effectively utilised as diagnostic markers in a clinical setting. The primary aim of this systematic review was to review studies investigating changes in miRNA expression in clinical studies of human AKI. This review aims to consolidate current knowledge on miRNAs as diagnostic markers for AKI, providing insights to inform future research or clinical trial design.

## 2. Methods

This systematic review was performed according to the methods described in the Cochrane Handbook for Systematic Reviews of Interventions (Version 6.4) [28]. The study was reported according to the Preferred Reporting Items for Systematic Reviews and Meta-Analysis (PRISMA) guidelines [29]. The study protocol has been registered at PROSPERO (CRD42024520999).

### 2.1. Eligibility Criteria

Studies reporting miRNA expression in patients diagnosed with AKI as a result of ischaemia–reperfusion injury, nephrotoxins, radiocontrast, shock, trauma, sepsis, or cardiopulmonary bypass were included. Studies were published and written in English and included observational trials, irrespective of blinding, date of publication, and sample size. Editorials, review articles, randomised control trials, in vitro and animal models, and studies reporting previously published data were excluded.

### 2.2. Information Sources

Eligible studies were identified by searching Pubmed, Cochrane Library, Scopus, and Ovid Medline from inception until February 2024 using the following terms: (ischaemia OR ischemia OR ischaemic OR ischemic OR ischaemia reperfusion OR ischemia reperfusion OR shock OR trauma OR sepsis OR nephrotoxicity OR radiocontrast OR cardiopulmonary bypass OR cardiac surgery) AND (kidney OR renal OR kidney injury OR renal injury) AND (miRNA OR mi-RNA OR microRNA OR micro-RNA OR miR OR mi-R) AND human NOT cancer]. The reference lists of all relevant review articles identified during the search were manually screened by two authors (NB, MR) and added as additional records identified through other sources. The full search strategy is detailed in Appendix A.

### 2.3. Study Selection 

Relevant studies were managed using EndNote 21.2 (Clarivate, Philadelphia, PA, USA). Three reviewers (NB, DM and MR) independently selected studies for inclusion according to the predetermined inclusion and exclusion criteria. Disagreements were resolved through discussion and adjudicated where required by MW. Following the removal of duplicates, study abstracts were assessed and excluded if they met the following criteria: (1) study was a review, editorial or conference abstract, (2) study was a randomised control trial, (3) study included previously published data, (4) study was an in-vitro model, (5) study was conducted on animals, (5) study was not reporting AKI, (6) study reported AKI in the context of kidney transplant, (7) study was reporting chronic kidney disease (CKD), or (8) study was reporting diabetic kidney disease (DKD). To control for condition-specific miRNA expression patterns associated with oncogenesis rather than AKI-related mechanisms, abstracts were further excluded if: (9) study was performed in patients with cancer. Following full-text screening, studies were excluded if: (1) miRNA was not a measured outcome, (2) the study was not reporting AKI, (3) there was no comparison of miRNA between AKI and non-AKI control, (4) the article was not in English, or (5) miRNA could not be quantified. The inclusion criteria were: (1) study was performed in human samples derived from blood, urine or kidney tissue, (2) miRNA was measured using PCR, array or sequencing, or (3) study was reporting AKI occurring as a result of ischaemia–reperfusion injury, sepsis, shock, nephrotoxicity, radiocontrast, or cardiopulmonary bypass surgery.

### 2.4. Data Extraction

Data extraction was performed by DM and NB using a standardised proforma as follows: author, study title, journal, year of publication, study design, AKI assessment criteria, potential cause of AKI, time point, age, miRNA expression, sample type, method of miRNA quantification, measurement units and PCR reference controls. For each measured variable, the mean or median and the standard deviation (SD), standard error (SEM), or interquartile range (IQR) were extracted. The number of subjects in AKI and non-AKI groups was recorded. Numerical data presented only graphically and not within the text was extracted using WebPlot Digitizer 4.7 [30].

### 2.5. Risk of Bias

The risk of bias and study quality were independently evaluated by two reviewers (NB and MR) using the adjusted Newcastle–Ottawa Quality Assessment Scale (NOS) for case-control and cross-sectional studies (File S1). Studies were evaluated based on three perspectives: the selection of study groups, the comparability of the groups, and the ascertainment of either the exposure or outcome of interest for case-control or cross-sectional studies, respectively. Case-control study quality was scored as poor (0–3 points), fair (4–5 points), good (6–7 points), or very good (8–9 points), whilst cross-sectional study quality was scored as poor (0–4 points), fair (5–6 points), good (7–8 points), or very good (9–10 points). Disagreements between reviewers were resolved by discussion and adjudicated by MW.

### 2.6. Data Synthesis

The analysis was conducted and graphics prepared using R version 4.4.0 [31]. Medians and SEM or IQR were transformed to means and SD using the Box-Cox algorithm and R package estmeansd [32]. Meta-analysis was performed with metafor (version 4.6-0) R package [33]. Random-effects models were fitted with the restricted maximum-likelihood (REML) estimator using rma function. Heterogeneity was assessed using the Cochran Q test [34]. The I^2^ statistic was used to estimate the proportion of variability across studies that is attributed to heterogeneity, rather than to sampling error. Publication bias was assessed using Egger’s test [35]. Results are presented as Standardised mean difference (SMD) with (95% confidence intervals) and *p* values.

## 3. Results

The electronic searches identified 1698 records and the reference list screening identified an additional four records. After the initial screening and removal of duplicates (n = 656), reviews (n = 395), and other titles outside the scope of this review (n = 536), 115 studies were assessed for eligibility. A total of 44 studies were excluded because they did not measure miRNAs (n = 17), did not report AKI (n = 13), or did not include AKI control groups (n = 9). Moreover, six papers were not in English, and in three cases, we could not quantify the measured miRNA from immunohistochemistry images or heatmaps. The eligibility screening resulted in 71 included studies (Figure 1) [26,36,37,38,39,40,41,42,43,44,45,46,47,48,49,50,51,52,53,54,55,56,57,58,59,60,61,62,63,64,65,66,67,68,69,70,71,72,73,74,75,76,77,78,79,80,81,82,83,84,85,86,87,88,89,90,91,92,93,94,95,96,97,98,99,100,101,102,103,104,105]. The characteristics of all included studies are in Appendix A.

The included studies were published between 2012 and 2024 and investigated AKI in patients with sepsis (44 studies), undergoing cardiac surgery (10 studies), ICU patients (7 studies), patients subjected to nephrotoxic agents (4 studies) and patients with ischaemia (1 study). Nine studies did not specify the cohort characteristics. Aguado-Fraile et al. [36] investigated ICU, cardiac surgery, sepsis and patients with unspecified conditions; Wang et al. [84] examined patients with sepsis and unspecified conditions. The miRNAs were measured in serum (51 studies), urine (17 studies), plasma (8 studies), urinary exosomes (1 study), peripheral blood mononucleated cells (PBMC), blood, endothelial cells and blood RNA (1 study, each). Nine studies measured miRNA in more than one sample type. The measurement times were related either to the diagnosis of AKI (patients with sepsis, ischaemia or ICU cohorts) or to the application of an insult (nephrotoxic agent or surgery).

### 3.1. Assessment of Methodological Quality

Of the 71 studies included in this review, 58 included cross-sectional cohorts, 11 contained case-control cohorts, and 2 included both case-control and cross-sectional cohorts. The evaluation of methodological quality for each study is reported in Appendix A. Using the NOS scale, only one study was found to be without methodological limitation ([38], Figure 2; Appendix A). The most common limitation was the representativeness of the sample (93.5%) or cases (71.4%). The mean NOS scores were 5.7 (fair) and 7.1 (good) for cross-sectional and case-control cohorts, respectively.

### 3.2. Untargeted Assays

MiRNAs were measured in cardiac surgery and sepsis cohorts (two studies each); nephrotoxicity was investigated in one study, and two studies did not specify the cohort characteristics. The AKI assessment criteria varied between studies. AKI was assessed using three different criteria: Kidney Disease Improving Global Outcomes (KDIGO, three studies), Acute Kidney Injury Network (AKIN, three studies) and Risk, Injury, and Failure; and Loss; and End-stage kidney disease (RIFLE, one study). Five studies measured miRNA in serum, two in urine, and one in plasma. Shihana et al. [74] measured miRNA in serum and urine. The untargeted methods of miRNA assessment included PCR arrays (four studies), hybridisation arrays (one study), and next-generation sequencing (one study). The miRNAs were measured at AKI diagnosis or after the AKI diagnosis (two studies, each). Two studies used untargeted assays in pre-operative plasma of cardiac surgery patients, and one study used them at 4, 8, 16, 16, 24 h, and 3 months after the application of a nephrotoxic contrast agent. Sullo et al. [75], Miller et al. [26], and Shihana et al. [74] measured miRNAs at more than one time point. Levels of miRNA were reported as fold changes or log fold changes between AKI and control groups. The untargeted assays identified 856 miRNAs differentially regulated in AKI patients. Twelve of them had non-standard names, unrecognised by miRbase [106]. None of the significant miRNAs were identified by more than one study. Appendix A shows extracted values for significant miRNAs measured using untargeted assays.

### 3.3. Targeted Assays

The investigated cohorts included patients with sepsis (43 studies), cardiac surgery patients (9 studies), ICU patients (7 studies), patients who received nephrotoxic agents (3 studies), and patients with ischaemia (1 study). Eight studies did not specify cohort characteristics. Aguado-Fraile et al. [36] investigated miRNAs in cardiac surgery and ICU patients, Wei et al. [85] in cardiac surgery patients and patients with undefined characteristics, and Wang et al. [84] in patients with and without sepsis. KDIGO was most commonly used to assess AKI (19 studies), followed by AKIN, changes in creatinine levels (7 studies, each) and RIFLE (2 studies). Four studies used a combination of KDIGO and AKIN, and one used a combination of RIFLE and AKIN to diagnose AKI. Thirty studies did not specify the AKI assessment method. Aguado-Fraile et al. [36] used AKIN, RIFLE, and changes in creatinine levels to assess AKI in three cohorts. The miRNAs were most commonly measured in serum and plasma (56 studies), followed by urine or urinary exosomes (17 studies). Other sample types included blood RNAs, circulating endothelial cells, kidney biopsy, and PBMCs (1 study each). All studies used qRT-PCR to assess the miRNA levels. The expression levels were reported as concentration values (2^−ΔΔCT^—48 studies, 2^−ΔCT^—9 studies, ΔCT—1 study, and fold changes—2 studies) relative to spike-in controls or endogenous miRNAs and mRNAs, whose levels did not change in AKI. In nine cases, it was unclear which measurement units were used. The most commonly used reference controls were U6 (42 studies) and cel-miR-39 (13 studies). Furthermore, nine studies did not specify the measurement units, and five studies did not disclose the reference controls (Figure 3A).

The studies that measured the miRNAs at time points related to AKI diagnosis most commonly assessed the miRNA levels after the AKI diagnosis (either unspecified or 24 h after diagnosis—41 studies, Figure 3B). Fourteen studies evaluated the miRNAs at the time of the AKI diagnosis and three before the diagnosis. Studies that timed the measurements in relation to an insult (application of nephrotoxic agents or cardiovascular surgery) most commonly reported miRNA levels before the insult (eight studies) or 24 h after (seven studies, Figure 3B).

Out of 109 analysed miRNAs, only miR-495-3p, miR-370-3p, and miR-21 were measured in two or more independent studies (Appendix A). MiR-495-3p and miR-370-3p were significantly downregulated after AKI diagnosis in serum or plasma from patients with sepsis (each in two independent studies). Both miRNAs were also measured in the urine of patients with sepsis, each in one study, where their levels were also significantly lower in AKI patients when measured after the diagnosis (Figure 4A).

MiR-21 was significantly upregulated in the urine of AKI patients when measured in ICU patients and two unspecific cohorts. The meta-analysis model results indicate a statistically significant effect size of 0.56, with no evidence of heterogeneity among the included studies and no significant publication bias (Figure 4B). The evidence for differential regulation of miR-21 in cardiac surgery patients with AKI when measured at 24 or 19 h is less clear. The meta-analysis revealed a non-significant effect size with substantial heterogeneity among the included studies. The high I^2^ value indicates considerable variability that cannot be attributed to sampling error alone. There was also significant publication bias, as indicated by the Egger’s test *p*-value < 0.05 (Figure 4C).

Two more independent studies measured MiR-21 in urine in samples from patients before cardiac surgery and 6 and 12 h afterward. Another study measured miR-21 after exposure to nephrotoxic agents. As before, the results are variable. The effect sizes of miR-21 in plasma or serum are also variable in patients undergoing cardiac surgery or with sepsis (Figure 4D).

## 4. Discussion

The systematic review identified 71 studies investigating the role of miRNAs in acute kidney injury across various patient cohorts spanning over a decade of research. There was considerable variability in the employed AKI assessment criteria, analysed sample types, reporting units, times of measurements, and used controls. Only three miRNAs were reported in more than one study, with miR-495-3p and miR-370-3p showing similar effects and miR-21 varying between cohorts and measurement times.

### 4.1. Clinical Significance

MiRNAs measured in more than one study included miR-495-3p and miR-370-3p, both were downregulated in sepsis patients with AKI. However, the measurements were performed after the AKI diagnosis, disqualifying the miRNAs as potential markers of AKI. However, they could shed light on potential mechanisms of AKI in patients with sepsis. Both miRNAs were previously shown to inhibit inflammatory responses. MiR-495-3p inhibited LPS-induction of NLRP3 inflammasome in vitro and vivo [107], and miR-370-3p can modulate inflammatory responses during epithelial-mesenchymal transition in colorectal cancer [108].

Another miRNA tested in multiple studies is miR-21. It is one of the best-investigated miRNAs, with more than 7000 published studies spanning conditions and processes, such as cancer, cardiovascular diseases, inflammation and immune response, liver function, or fibrosis. In this study, we were able to analyse data from only two sets of three independent studies reporting the levels of miR-21 in AKI. Three studies measured miR-21 in urine samples at the time of the AKI diagnosis, and the results showed consistent upregulation. This was not evident 19–24 h after cardiovascular surgery when levels of miR-21 were either increasing or decreasing in AKI patients. The role of miR-21 in AKI was investigated in 14 animal studies evaluated by Zankar et al. [27], who showed that miR-21 antagonism was associated with increased kidney injury. The increased levels of miR-21 in human studies potentially contradict these findings. However, urinary and circulating levels of miR-21 may behave differently during AKI.

### 4.2. Strengths and Limitations 

To our knowledge, this is the first systematic review of miRNA levels in AKI in human studies. We used comprehensive search strategies incorporating a range of data sources and widely accepted contemporary methods for assessing methodological quality and statistical analysis.

The major limitation of this study is the lack of reproducibility in miRNA reporting between included studies. Despite the potential thousands of tested miRNAs and 856 identified as significant in AKI, no single miRNA was reported in more than one untargeted study. This may be attributed to the diverse underlying mechanisms of AKI [109]. There was great variability among cohorts selected for the untargeted assays. The two studies in cardiac surgery patients were performed in adult and paediatric populations, and the two studies in patients with sepsis used different measurement techniques (TaqMan arrays vs. hybridisation arrays) and assessed AKI using KDIGO and AKIN criteria, which vary in the diagnosis [110]. A similar situation was evident in the results of targeted assays. Out of 109 miRNAs, only three were measured in more than one study.

The targeted assays lacked consistency in the selection of control and included healthy individuals or patients with similar characteristics without AKI. Moreover, all studies reported relative concentration values instead of copy numbers. Such data heavily rely on the selection of the reference control, which included endogenous controls or spike-in miRNAs like cel-miR-39 or synthetic oligos. The most commonly used reference control was U6, which was shown to be an unsuitable control for miRNA quantification in circulation [111], particularly in patients with sepsis [112]. There was also a lack of consistency in how the expression levels were reported. It included ΔCT and ΔΔCT values; in some cases, they were normalised against controls or urinary creatinine.

The assessment of methodological quality identified further limitations. The NOS scale indicated that out of the 71 studies included in this review, only one study was without methodological limitation. Furthermore, most studies failed to recruit patients that were representative of the AKI population, indicating that the majority of studies were at risk of selection bias.

The methodological limitations of our review include the exclusion of randomised control trials. The inclusion could introduce bias related to intervention effects and increase heterogeneity due to different designs compared with diagnostic studies. We did not contact study authors for unpublished or missing data, or when it was not possible to extract numerical data from immunohistochemistry images or heatmaps. Finally, we decided not to include studies performed in patients with cancer due to its strong documented effect on circulating miRNA profiles [113,114,115]. Consequently, our study design may have missed some data, potentially limiting the comprehensiveness of our review.

## 5. Conclusions and Future Work

In conclusion, our review reveals a significant knowledge gap in the utility of miRNAs as diagnostic markers for AKI. The lack of reproducibility in miRNA reporting may be attributed to both the diverse mechanisms of AKI and the considerable heterogeneity and variability among the included studies. This highlights the need to standardise AKI diagnosis, experimental design, and miRNA reporting to improve comparability across studies. Future research should employ validated AKI diagnostic criteria and consistent methods of miRNA quantification, including standardised reference controls and units of measurement. Alternatively, absolute concentration units should be used instead of relative. Addressing the limitations identified in this review is essential for gaining a better understanding of miRNAs in AKI and their applicability as predictive biomarkers.

## Figures and Tables

**Figure 1 biomedicines-12-01695-f001:**
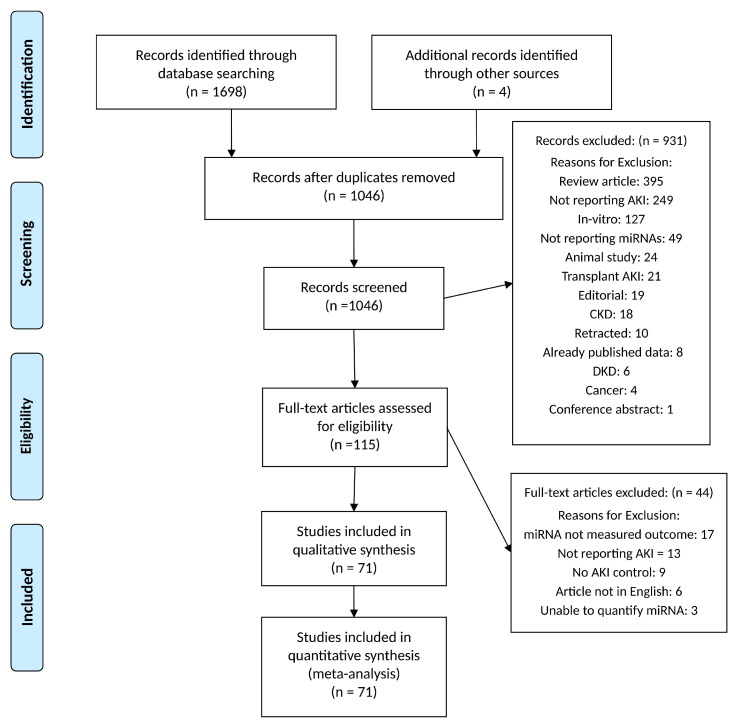
PRISMA flow diagram for study selection [29].

**Figure 2 biomedicines-12-01695-f002:**
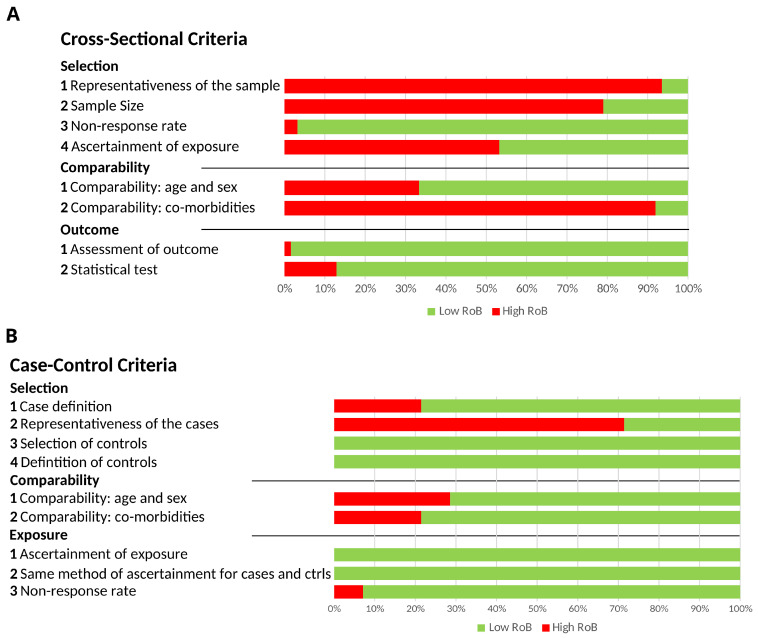
Assessment of study bias—Newcastle Ottawa-Scale (NOS) adapted for (**A**) cross-sectional and (**B**) case-control studies.

**Figure 3 biomedicines-12-01695-f003:**
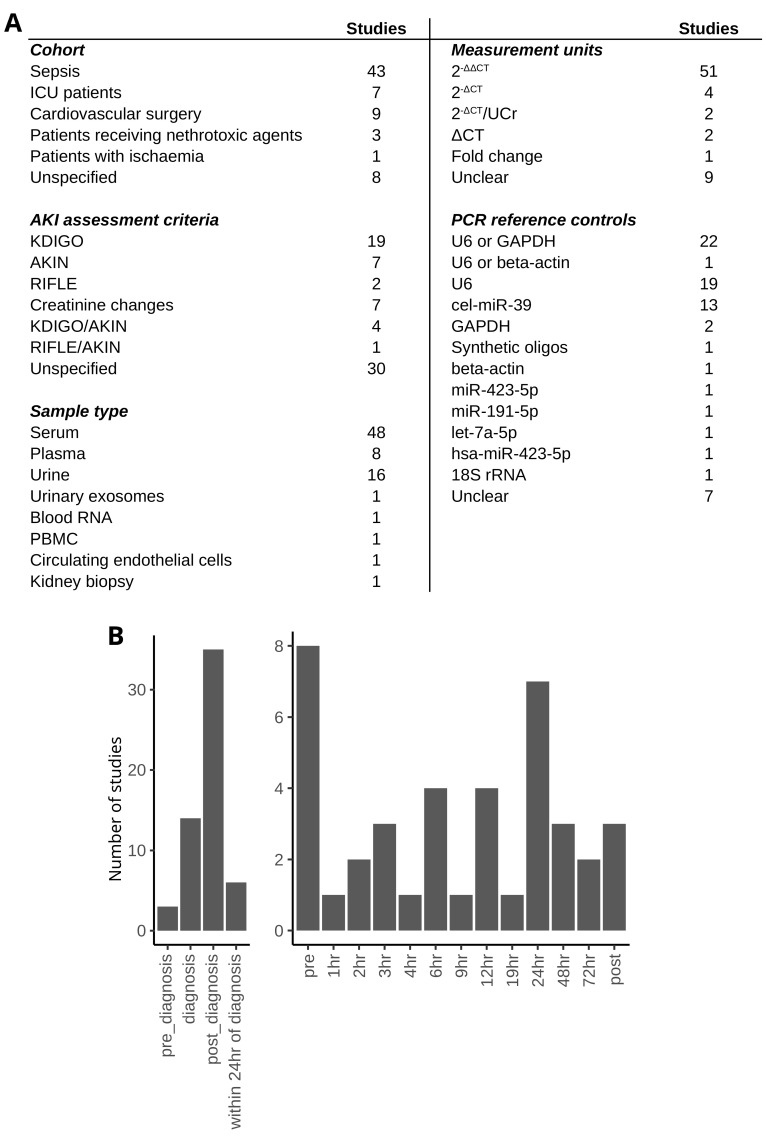
Summary of included studies using targeted assays—(**A**) Study characteristics. (**B**) Number of studies per time point.

**Figure 4 biomedicines-12-01695-f004:**
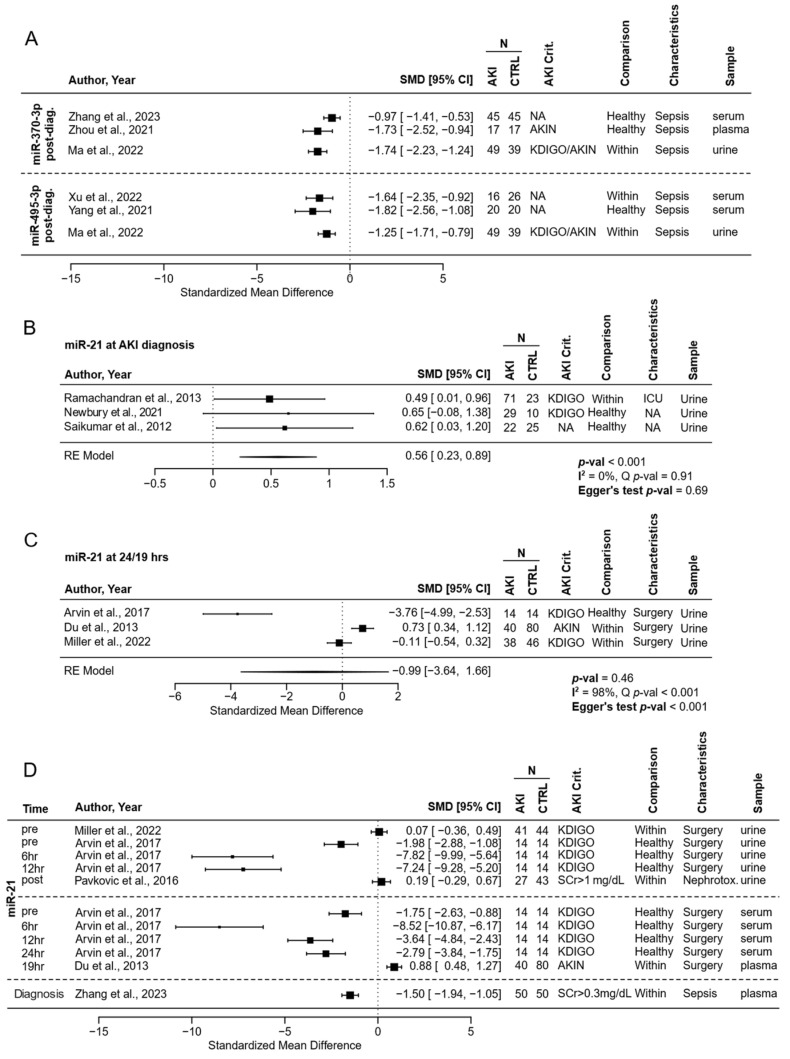
Differentially expressed miRNAs across studies—(**A**) Standardised mean differences of miR-370-3- and miR-494-3p post-diagnosis in serum, plasma, and urine between controls and patients with AKI. (**B**,**C**) Forest plots for miR-21 across studies at the time of AKI diagnosis (**B**) and 19 or 24 h after cardiac surgery (**C**). (**D**) Standardised mean differences of miR-21 in urine, plasma, or serum in patients after cardiac surgery, receiving nephrotoxic agents, or with sepsis. N—number of patients; SMD, standardised mean difference; CI, confidence interval; RE, Random Effect. The comparison indicates whether the AKI patients were compared with healthy controls (Healthy) or patients with similar clinical characteristics (Within). AKI Crit. indicates the criteria used to diagnose AKI [26,38,41,67,69,70,71,72,87,90,95,96,104].

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
