# Peer review of "A Systematic Review and Meta-Analysis of MicroRNA as Predictive Biomarkers of Acute Kidney Injury"

_biomedicines, 2024, doi:10.3390/biomedicines12081695_

Round 1

Reviewer 1 Report

Comments and Suggestions for Authors

The manuscript is well-structured and presents a thorough review of MicroRNA as predictive biomarkers of Acute Kidney Injury. Here are several major comments which needs to be addressed in order to assure the highest quality of the presented data.

1.     The language is generally precise. However, consider a thorough proofreading to correct any minor grammatical errors or typos. For example, Line 14: “This review of studies measuring miRNAs in AKI aimed to verify miRNAs as diagnostic markers." should be "This review of studies measuring miRNAs in AKI aims to verify miRNAs as diagnostic markers."

2.     The introduction is not rich enough. it is better to include more recent references to support the significance of AKI and advancements in biomarker research. And the specific challenges in current diagnostic methods for AKI are necessary to elaborate.

3.     Provide more detail on the data extraction process, particularly how disagreements were resolved.

     4.     Expand on the limitations of the review and how they might affect the findings. manuscript is well-structured and presents a t

Comments on the Quality of English Language

The language is generally clear . However, the language needs to be revised by native speakers to correct any minor grammatical errors or typos.

Author Response

The manuscript is well-structured and presents a thorough review of MicroRNA as predictive biomarkers of Acute Kidney Injury. Here are several major comments which needs to be addressed in order to assure the highest quality of the presented data.

  1. The language is generally precise. However, consider a thorough proofreading to correct any minor grammatical errors or typos. For example, Line 14: “This review of studies measuring miRNAs in AKI aimed to verify miRNAs as diagnostic markers." should be "This review of studies measuring miRNAs in AKI aims to verify miRNAs as diagnostic markers."

Answer:

Thank you for acknowledging the strength of our review. We have conducted a thorough proofreading of the document and corrected grammatical errors and typos which are highlighted throughout.

  1. The introduction is not rich enough. it is better to include more recent references to support the significance of AKI and advancements in biomarker research. And the specific challenges in current diagnostic methods for AKI are necessary to elaborate.

Answer:

We agree with the reviewer and improved the introduction of our manuscript accordingly. We have incorporated references of recent biomarkers (TIMP2/IGFBP7 and Dickkopf) that support the recent advances in biomarker research. Additionally, we have elaborated on the specific challenges in current diagnostic methods for AKI, which include the lack of specificity and sensitivity of serum creatinine and urine output which leads to the inaccurate and delayed diagnosis of AKI (Lines 37-42).

  1. Provide more detail on the data extraction process, particularly how disagreements were resolved.

Answer:

Our manuscript details on Line 77 that disagreements were resolved by discussion between three reviewers and adjudicated where necessary by a fourth reviewer. We believe that this is in accordance with the Cochrane Handbook for Systematic Reviews of Interventions that details “…assessors may attempt to resolve disagreements via discussion, and if that fails, call on another author to adjudicate the final judgement.” [1]. We have improved the coherence of this sentence to ensure clarity to the reader regarding study selection disagreements.

  1. Expand on the limitations of the review and how they might affect the findings.

Answer:

We agree with the reviewer that exploring limitations of our study design is imperative for evaluating the overall findings of our review. We have now detailed how both excluding randomised control trials and not contacting study authors for missing data may result in overlooked data potentially leading to a less comprehensive analysis (Lines 281-288). 

References

1. Higgins, J. Cochrane handbook for systematic reviews of interventions. http://www. cochrane-handbook. org 2008.

Reviewer 2 Report

Comments and Suggestions for Authors

The manuscript offers a thorough evaluation of the potential of microRNAs as biomarkers in acute kidney injury. This paper would be the first meta-analysis on this specific topic.

The authors present their data in a clear and systematic way. The paper is informative and easy to read.

I would only avoid repeating the data from figures 1 and 3 in the text.

Author Response

The manuscript offers a thorough evaluation of the potential of microRNAs as biomarkers in acute kidney injury. This paper would be the first meta-analysis on this specific topic.

The authors present their data in a clear and systematic way. The paper is informative and easy to read.

I would only avoid repeating the data from figures 1 and 3 in the text.

Answer:

Thank you for acknowledging the importance of our review. We agree that there is a repetitive nature of how the data from Figure 1 and 3 are presented in the text. We have removed Line 130-141 and adjusted Line 153-156 accordingly to ensure the summary of the diagnostic criteria used to define AKI is not repeated.

Reviewer 3 Report

Comments and Suggestions for Authors

Naomi Brown and collaborators have carried out a review in which they aim to test the usefulness of different microRNAs in the diagnosis of acute kidney injury of various origins. I think this is a very interesting topic, since plasma creatinine (the most widely used parameter to date) is a late biomarker that is only altered when the kidney has been severely damaged, so the identification of other biomarkers with greater sensitivity is of great interest in the field of nephrology. The main conclusion of this work is that more research is still needed in this field, as the types of microRNAs evaluated are very different, there is a lot of heterogeneity and the sampling times and biological matrices used are also very different (this has not allowed a very complete meta-analytical work to be carried out). However, the methodology applied is correct and rigorous, as is the presentation of the results.

I can only point out some minor aspects to be revised:

- What is the reason for including "NOT cancer" in the search strategy? I interpret that it is possible that this pathology can alter the levels of microRNAs, but I think the reason should be described in more detail in the text (as they are probably missing a lot of work on cisplatin-induced AKI, for example).

- Line 74: the text states "(2) study was a randomised control trial", what does control trial mean? does it mean a clinical trial in which a drug or intervention is evaluated against a control group? it should be better described.

- Line 89: what does "mode of AKI" mean?

- Lines 95-98: what was the cut-off point applied for the Newcastle-Ottawa scale to decide whether a paper was of good quality or not?

- I think Table S2 is important enough to be included in the main document, not as supplementary material.

- Why are the references of the 71 papers included in the study not included in the References section?

Author Response

Naomi Brown and collaborators have carried out a review in which they aim to test the usefulness of different microRNAs in the diagnosis of acute kidney injury of various origins. I think this is a very interesting topic, since plasma creatinine (the most widely used parameter to date) is a late biomarker that is only altered when the kidney has been severely damaged, so the identification of other biomarkers with greater sensitivity is of great interest in the field of nephrology. The main conclusion of this work is that more research is still needed in this field, as the types of microRNAs evaluated are very different, there is a lot of heterogeneity and the sampling times and biological matrices used are also very different (this has not allowed a very complete meta-analytical work to be carried out). However, the methodology applied is correct and rigorous, as is the presentation of the results.

I can only point out some minor aspects to be revised

  1. What is the reason for including "NOT cancer" in the search strategy? I interpret that it is possible that this pathology can alter the levels of microRNAs, but I think the reason should be described in more detail in the text (as they are probably missing a lot of work on cisplatin-induced AKI, for example).

Answer:

Thank you for highlighting this important aspect of our search strategy. We agree that the justification for not including cancer as an imposing miRNA-expression pattern should be explained within the text. We added justification in the methods section (Line 83-86) and mention it as a possible limitation in the discussion (Line 285-288). 

  1. Line 74: the text states "(2) study was a randomised control trial", what does control trial mean? does it mean a clinical trial in which a drug or intervention is evaluated against a control group? it should be better described.

Answer:

A randomised control trial is defined by NICE as “A study in which a number of similar people are randomly assigned to 2 (or more) groups to test a specific drug, treatment or other intervention.” [1]

  1. Line 89: what does "mode of AKI" mean?

Answer:

Thank you for drawing our attention to that term, we replaced it with “potential cause of AKI”.

  1. Lines 95-98: what was the cut-off point applied for the Newcastle-Ottawa scale to decide whether a paper was of good quality or not?

Answer:

We agree with the reviewer and the following proforma has been added to the methods section: “Case-control study quality was scored as poor (0-3 points), fair (4-5 points), good (6-7 points) or very good (8-9 points), whilst cross-sectional study quality was scored as poor (0-4 points), fair (5-6 points), good (7-8 points) or very good (9-10 points).” (Line 104-107). 

  1. I think Table S2 is important enough to be included in the main document, not as supplementary material.

Answer:

We agree with the reviewer, however, due to the size of Table S2 (4000 words) and word restrictions as per instruction to authors, we decided not to include it the main document. This will however be accessible in the Supplement file.

  1. Why are the references of the 71 papers included in the study not included in the References section?

Answer:

We agree with the reviewer that the 71 papers included in the study should be included in the reference section and have added these accordingly.